# Thermal management towards ultra-bright and stable perovskite nanocrystal-based pure red light-emitting diodes

Hongjin Li[1,2,4], Xiaofang Zhu[1,2,4], Dingshuo Zhang [1,2], Yun Gao [1,2], Yifeng Feng[1,2], Zichao Ma[1,2], Jingyun Huang [1,2], Haiping He [1,2,3], Zhizhen Ye [1,2,3] ✉ & Xingliang Dai [1,2,3] ✉

Despite the promising candidacy of perovskite nanocrystals for light-emitting diodes, their pure red electroluminescence is hindered by low saturated luminance, severe external quantum efficiency roll-off, and inferior operational stability. Here, we report ultra-bright and stable pure red light-emitting diodes by manipulating Joule heat generation in the nanocrystal emissive layer and thermal management within the device. Diphenylphosphoryl azide-mediated regulation of the nanocrystal surface synergistically enhances the optical properties and carrier transport of the emissive layer, enabling reduced Joule heat generation and thus lowering the working temperature. These merits inhibit ion migration of the $CsPb(Br/I)_3$ nanocrystal film, promising excellent spectra stability. Combined with the highly thermal-conductive sapphire substrates and implementation of pulse-driving mode, the pure red light-emitting diodes exhibit an ultra-bright luminance of 390,000 cd m$^{-2}$, a peak external quantum efficiency of 25%, suppressed efficiency roll-off, an operational half-life of 20 hours, and superior spectral stability within 15 A cm$^{-2}$.

Perovskite light-emitting diodes (PeLEDs) are emerging as promising candidates in display technology, addressing the demand for electroluminescence with high color purity, low power consumption, and cost-effective fabrication[1–6]. Pure red PeLEDs with emission of 620–650 nm, as one of the important components for the ultra-high-definition color gamut of the display, have achieved tremendous progress on the external quantum efficiency (EQE) of devices defined by the ratio of extracted photons over injected charges[7,8]. Despite peak EQEs having exceeded 20%, critical challenges remain, including low saturated luminance, severe EQE roll-off, and inferior operational stability, impeding their further practical application[9–11]. Most pure red PeLEDs present a maximum luminance of thousands of nits, and inadequate performance at a display brightness of 1000 nits.

Insufficient Joule heat dissipation, originating from the high resistance of the emissive layers and low thermal conductivity of the glass substrate, is considered one of the critical factors leading to detrimental performance on PeLEDs[12–16]. Heat accumulation would increase thermally activated trap sites and accelerate ion-related processes, leading to thermal degradation or decomposition of perovskite emissive layer[12–15]. The elevated temperatures may further disturb the charge balance and influence carrier recombination in PeLEDs[16].

Several thermal management strategies have been developed, encompassing two major approaches: accelerating the thermal dissipation capacity of operating devices by attaching heat spreaders or utilizing high-thermal-conductivity substrates such as silica or sapphire[12,17]; and suppressing the generation of Joule heat by

[1]School of Materials Science and Engineering, State Key Laboratory of Silicon and Advanced Semiconductor Materials, Zhejiang University, Hangzhou 310027, P. R. China. [2]Wenzhou Key Laboratory of Novel Optoelectronic and Nano Materials and Engineering Research Centre of Zhejiang Province, Institute of Wenzhou, Zhejiang University, Wenzhou 325006, P. R. China. [3]Shanxi-Zheda Institute of Advanced Materials and Chemical Engineering, Taiyuan 030002, P. R. China. [4]These authors contributed equally: Hongjin Li, Xiaofang Zhu. ✉e-mail: yezz@zju.edu.cn; shanfeng@zju.edu.cn

addressing internal factors, such as doping charge-transport layers[18,19] and optimizing light extraction nanostructure geometry[14,15]. Among them, enhancing the electrical and thermal conductivity of the emissive layers and minimizing nonradiative recombination losses contribute significantly to alleviating the accumulated heat[12,20]. Surface chemistry plays essential roles in determining the conductivity[21,22] and regulating surface defects of perovskite materials[23,24], exerting a notable influence on the performance of the devices. Previous investigations have revealed that ligand manipulation can increase the thermal conductivity of nanocrystal (NC) films by 6-10 times[22]. Inorganic potassium iodide ligands with a high-thermal conductivity demonstrated enhanced heat dissipation of NC films, achieving a peak EQE over 23%[25]. Recently, small-sized aromatic tryptophan ligands were employed to coordinate with CsPb(Br/I)₃ NCs[26], exhibiting less-detective surface and superior charge-transport properties of the assembled emissive layer, thereby enabling a pure red PeLED with a maximum luminance of 12,910 cd m⁻² and a peak EQE of 22.8%. Nevertheless, it remains an unprecedented challenge to achieve pure red PeLEDs with high efficiency and luminance, suppressed efficiency roll-off, and spectra stability.

In this work, we aim to synergistically suppress Joule heat generation and improve the thermal dissipation of the device to achieve highly bright, efficient, and stable pure red PeLEDs. Mixed-halide CsPb(Br/I)₃ NCs are considered promising emitters because of the excellent spectra tunability, the efficient radiative recombination, and the limited halide separation in a single particle. Conjugated diphenylphosphoryl azide (DPPA) is employed to regulate the anchoring ligands of CsPb(Br/I)₃ NCs during synthesis, which synergistically enhances the optical properties and carrier transport performance of

the NC film. Combined with a thermally conducting sapphire substrate and the pulse-driving mode, the overall electroluminescent performance of the NC film is investigated.

## Results and discussion

### Surface regulation of CsPb(Br/I)₃ nanocrystals

The surface of primitive CsPb(Br/I)₃ NCs dynamically coordinates with long-carbon-chain acid and amine ligands[27]. The weak binding nanocrystals always suffer from ligand stripping, leading to surface defects[28]. Furthermore, the insulating nature of these ligands results in poor electrical and thermal conductivity of the NC film[22,29]. To address these issues, we introduced DPPA into the synthesis to regulate the surface of the NCs, referred to as DPPA-NCs (see details in the Methods). The π-conjugated benzene ring is recognized for its superior conductivity compared with the carbon chain, and the phosphonate radical possesses a strong affinity with nanocrystals[30,31]. The surface regulation mechanisms of NC mediated by DPPA are schematically presented in Fig. 1a and Supplementary Fig. 1. Specifically, DPPA reacts with carboxylic acid to generate diphenyl phosphate (DPP), serving as an alternative ligand on the surface of NCs (Fig. 1b). The acyl azide intermediately further reacts with oleylamine[32] (Supplementary Fig. 1), yielding byproducts that can be washed away in subsequent purification steps. It is a process of anchoring aromatic ring ligands to the surface of NCs accompanied by consuming long-chain carboxylic acid and oleylamine. During the crystallization, the -RNH₃⁺ competes with Cs⁺ to terminate the surface of the NCs[33]. The consumption of oleylamine in the precursor by DPPA results in a reduction of -RNH₃⁺ and enables Cs⁺ to continue growing on the surface. Statistical analysis of more than 100 particles in transmission electron microscopy images

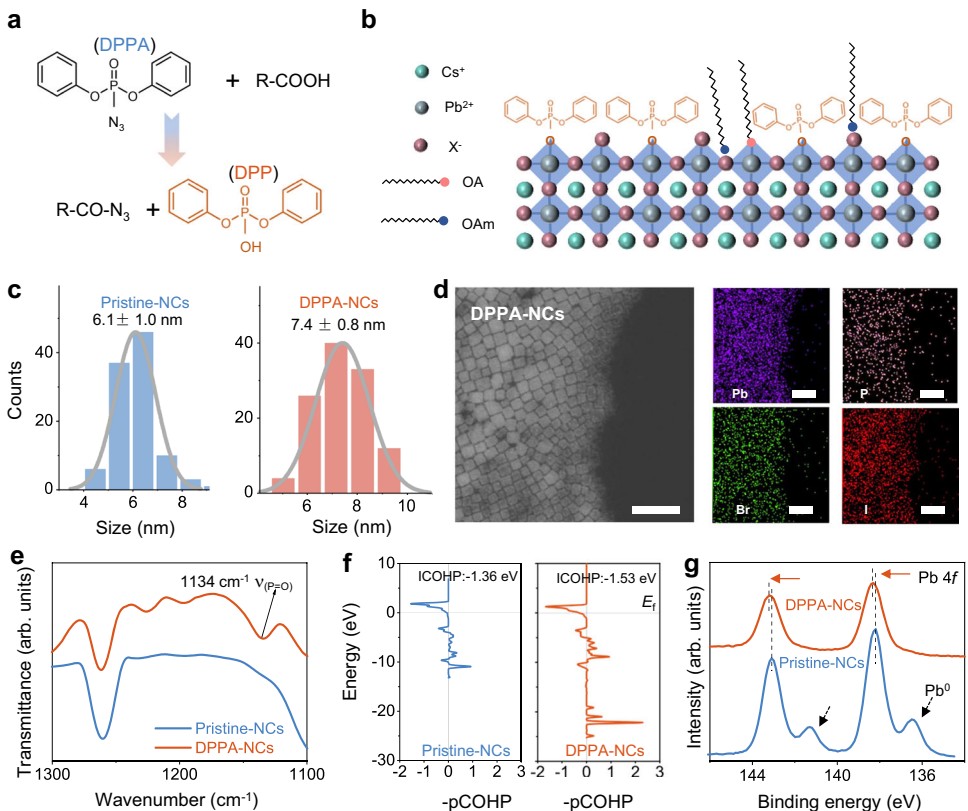

**Fig. 1 | Surface regulation of the CsPb(Br/I)₃ NCs. a** Schematic diagram of the reaction between DPPA with carboxylic acids. **b** Schematic illustration of the surface of DPPA-NCs. **c** Size distribution histograms of the nanocrystals. **d** HAADF-STEM image and corresponding elemental mapping images of the DPPA-NCs. Scale bar: 50 nm. **e** FTIR spectra of the NCs, showing a distinct bending vibration of the P=O bond in DPPA-NCs. **f** Crystal orbital Hamilton population (COHP) calculations for the Pb-O bonds between perovskite lattice and surface ligands for the pristine-NCs and DPPA-NCs, respectively. The left and right sides represent the antibonding and the bonding states, respectively. **g** High-resolution XPS spectra of Pb 4*f*.

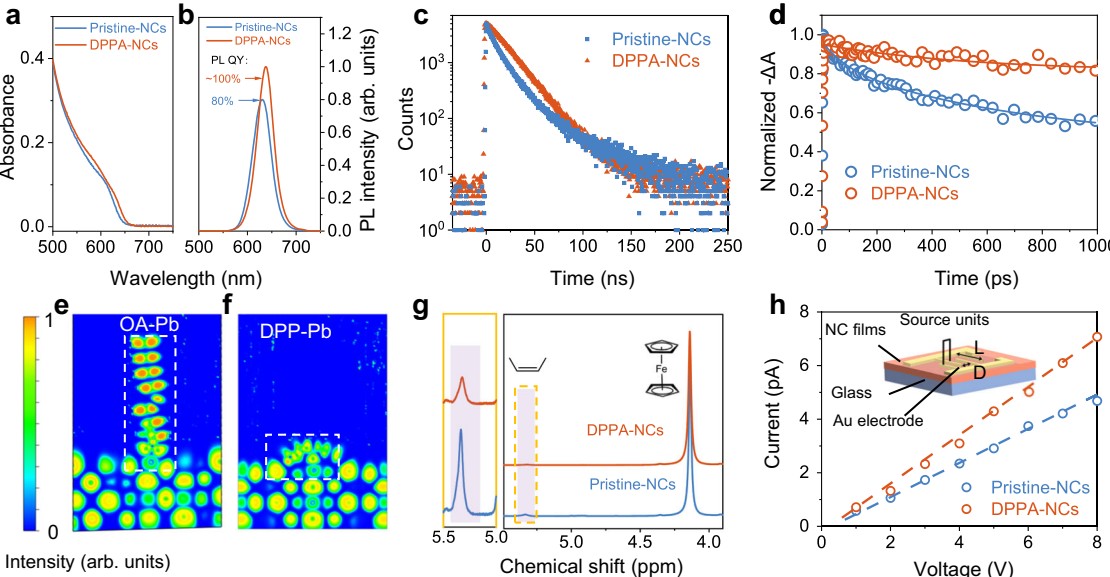

**Fig. 2 | Optoelectronic properties of the CsPb(Br/I)₃ NCs. a** UV-vis absorption spectra. **b** PL spectra. **c** Time-resolved photoluminescence decay curves of the NC in solution. **d** TA response of the NCs measured with a pump light wavelength at 500 nm. The probe light wavelength is determined by their maximum exciton bleach. Pump light intensity: 3 μJ cm⁻². **e, f** The electron localization function (ELF) of OA (**e**) and DPP (**f**) ligands on the CsPb(Br/I)₃ surfaces. The yellow indicates a high-value region and the green indicates a low-value region. **g** Quantitative ¹H NMR spectra of the NCs using ferrocene as the reference material. The left exhibits the enlarged spectra of the alkenyl region ($\delta = 5.2$–5.4 ppm). **h** Conductivity characterization of the NCs using a lateral device structure with a stripe length ($L$) of 7.4 mm and channel width ($D$) of 150 μm, as shown in the schematic diagram in the inset.

indicates a size distribution of $7.4 \pm 0.8$ nm of the DPPA-NCs, larger than $6.1 \pm 1.1$ nm of the pristine-NCs (Fig. 1c and Supplementary Fig. 2). The large size of the DPPA-NCs is favorable for charge carrier transport in NC films[34]. We note that directly adding DPP (labeled as DPP-NCs) cannot achieve the same positive effect as DPPA, as will be discussed later.

We investigated the surface and crystal structure of the CsPb(Br/I)₃ NCs regulated by DPPA-mediated synthesis. The cubic morphology of the pristine-NCs and DPPA-NCs is observed in high-angle annular dark-field scanning transmission electron microscopy (HAADF-STEM), as shown in Fig. 1d and Supplementary Fig. 3. Element mapping reveals a uniform spatial distribution of the phosphorus on the DPPA-NCs, suggesting the presence of DPP in the purified NCs. Besides, the Fourier-transform infrared (FTIR) spectrum of the DPPA-NCs shows distinct vibrational bonds of P=O stretching at 1134 cm⁻¹, referring to the features of DPP⁻ ligands (Fig. 1e and Supplementary Fig. 1b). These findings suggest that DPP is successfully anchored on the surface of DPPA-NCs (Fig. 1b). Both the pristine-NCs and DPPA-NCs exhibit strong X-ray diffraction (XRD) peaks at 14.1° and 28.8° (Supplementary Fig. 4), which match well with the α-phase of perovskite[35]. High-resolution transmission electron microscopy images (Supplementary Fig. 5) of both NCs exhibit the lattice fringes of ~0.30 nm corresponding to the (200) planes of the CsPb(Br/I)₃ NCs, indicating that the incorporation of DPPA into NCs has a negligible influence on the perovskite crystal structures.

To gain a comprehensive understanding of the bonding nature of the surface, we employed density functional theory (DFT) calculations to validate the distinction between DPP and carboxylic based on the optimized crystal structures (Supplementary Fig. 6). The crystal orbital Hamilton population (COHP) represents the local chemical-bonding properties[36]. The integral value of COHP (ICOHP) below the Fermi level reflects the number of bonding electrons shared between two atoms, providing a qualitative measure of bond strength. The ICOHP in this work is calculated between the surface Pb²⁺ of the NCs and O in deprotonated DPP⁻ and carboxylic, yielding values of −1.53 and −1.36 eV (Fig. 1f), respectively. The larger absolute value of ICOHP for DPPA-NCs represents the stronger bonding interactions between DPP⁻

and the surface of NCs. The DFT calculation results are consistent with the X-ray photoelectron spectroscopy (XPS) analysis. Compared with the pristine-NCs, the Pb $4f_{5/2}$ and $4f_{7/2}$ peaks display a -0.13 eV shift to higher binding energies for the DPPA-NCs (Fig. 1g), indicating lower electron cloud density around the Pb atom. The shift of the Pb peaks toward higher binding energies is evident for forming stronger ionic bonding between Pb²⁺ and DPP⁻ ligands[37]. Besides, the XPS core-level spectra of Pb $4f$ exhibit a distinct peak centered at 136.5 and 141.3 eV in the pristine-NCs, which are assigned to metallic Pb⁰ originating from the highly unsaturated Pb atom at the surface[26].

## Optoelectronic properties of the DPPA nanocrystals

The optical properties of the DPPA-NCs were investigated. Analogous ultraviolet-visible (UV-Vis) absorption spectra of the pristine-NCs and DPPA-NCs were observed, and the PL of NCs red-shifts from 630 to 638 nm after DPPA regulation (Fig. 2a, b). The red-shifted PL is consistent with the increased particle size of the DPPA-NCs. We then conducted nanosecond time-resolved photoluminescence (TRPL) and femtosecond transient absorption (fs-TA) measurements to unveil the carrier dynamics of the NCs. The TRPL spectra of the DPPA-NCs show a single exponential decay characteristic and a significantly prolonged lifetime ($\tau$) of 20.9 ns. In contrast, the TRPL of the pristine-NCs can only be fitted by bi-exponential decay along with the shortened effective lifetime of 11.8 ns and a much longer nonradiative recombination rate ($K_{nr}$) (Fig. 2c, Supplementary Fig. 7, and Supplementary Tables 1, 2). The carrier decay dynamics of the NCs in the picosecond time scale were further analyzed by the TA signal at the band-edge position. We tracked the change of $\Delta A$ at the ground state bleach as a function of delay time (Fig. 2d and Supplementary Fig. 8). The DPPA-NCs show a slow single-exponent kinetic bleach decay with a lifetime of 542 ps, while the pristine-NCs exhibit faster bleach decay (Supplementary Table 3). The above results indicate that the carriers remain longer at the band-edge instead of trapping to the defect states for the DPPA-NCs[38,39], verifying their less-defective surface feature and suppressed nonradiative recombination. As a result, the photoluminescence quantum yields (PLQYs) improve from 78% of the pristine-NC to near-

unity of the DPPA-NC in solution, and from 58% of the pristine-NC to 90% of the DPPA-NC in films. In addition, the DPPA-NC solution presents an overall improved PLQYs over a wide range of excitation intensities ranging from 0.5 to 100 mW cm$^{-2}$ (Supplementary Fig. 9). In general, the nonradiative recombination always leads to additional energy loss and contributes to heat generation[40]. The superior optical properties and diminished heat generation in the DPPA-NCs are favorable for LED performance.

Carrier transport performance of the NC films is another significant factor influencing Joule heat generation[41]. To reveal the carrier transport, the electron localization function (ELF) is a tool for studying the distribution and localization of electrons from different ligands on the NC surface[42]. The yellow region in the ELF signifies high values and localized charges, while the green region represents greater electron mobility (Fig. 2e, f). Consequently, the carboxylic exhibits pronounced localization characteristics throughout the entire long-chain regime. In contrast, DPP ligands demonstrate enhanced delocalization features, which proves advantageous for carrier transport in devices. Besides the electron localization of the ligands, the ligands content on the NC surface also plays a significant role in carrier transport. We conducted quantitative $^1$H nuclear magnetic resonance ($^1$H NMR) spectroscopy of the samples using ferrocene as an internal standard[7]. All $^1$H NMR signals from the NCs are normalized using the integral area of ferrocene resonance (4.19 ppm) as the reference. As depicted in Fig. 2g, the estimated integrated values of the alkenyl region ($\delta = 5.2$–$5.4$ ppm) are $8.57 \times 10^{-3}$ and $23.2 \times 10^{-3}$ for the DPPA-NCs and pristine-NCs, respectively. This signifies a reduction in the total concentration of long-chain carboxylic acid and oleylamine ligands for the DPPA-NCs, promoting carrier transport within the emissive layer. These merits enable the enhanced conductivity of the DPPA-NC film. The lateral devices based on DPPA-NC film demonstrate a higher current compared with the pristine-NC film (Fig. 2h).

Noteworthy, the direct addition of DPP does not effectively regulate surface ligands of the CsPb(Br/I)$_3$ NCs, as there is substantial competition from massive carboxylic acid and oleylamine ligands in the precursor for coordination on the surface without consumption. In the quantitative $^1$H NMR analysis, the decrement of alkene resonance in DPP-NCs is less than that in DPPA-NCs (Supplementary Fig. 10), inferring a lower substitution ratio of DPP against long-chain ligands.

We further assessed the thermal stability of these films, as shown in Supplementary Fig. 11. The PL intensity was recorded in situ as the temperature increased. The PL intensity of pristine-NC film decreases at 50 °C and retains only 50% of its initial efficiency when heated to 80 °C. In contrast, the DPPA-NC film shows a slower decline in PL intensity, maintaining 80% of its initial efficiency at 80 °C. Additionally, when the NC films are subjected to a hot plate at 80 °C, the DPPA-NC film retains above 50% of its initial PL intensity after 60 min of thermal stress, while the pristine-NC films drop below 40% within 20 min. This enhanced thermal stability of the DPPA-NC films is crucial in mitigating efficiency roll-off and ensuring the operational stability of the device.

## Electroluminescent performance of the DPPA nanocrystals

Having elucidated the beneficial effects of DPPA on the optical properties and conductivity of the CsPb(Br/I)$_3$ NCs, we are encouraged to evaluate the electroluminescent performance of the NC film. Light-emitting devices were fabricated with a multi-layered structure comprising indium tin oxide (ITO)/poly(3,4-ethylenedioxythiophene) polystyrene sulfonate: perfluorinated resin (PEDOT:PSS:PFI)/poly [bis(4-phenyl)(2,4,6-trimethylphenyl)amine](PTAA)/monolayer NC film/ 1,3,5-Tris(1-phenyl-1H-benzimidazol-2-yl)benzene (TPBi)/lithium fluoride (LiF)/Al (see Methods for details), as presented in Fig. 3a. The device incorporating a monolayer of NC emitter was designed to establish a short heat transfer pathway within the emissive layer[14]. The typical PeLEDs based on the DPPA-NCs exhibit a bright EL peak at 640 nm,

corresponding to CIE coordinates of (0.700, 0.298) (Supplementary Fig. 12). The flat-band energy level diagram of the pristine-NCs and DPPA-NCs is depicted in Supplementary Fig. 13, derived from the optical measurements and ultraviolet photoelectron spectroscopy analysis. The up-shifted energy level of the DPPA-NCs suggests a smaller hole-injection barrier compared with the pristine-NCs. Figure 3b illustrates the current density–voltage–luminance ($J$-$V$-$L$) curves of the two PeLEDs. The PeLED based on DPPA-NCs displays a lower turn-on voltage and higher luminance across the entire operational range, attributed to the enhanced optical properties and improved carrier transport. Specifically, the maximum luminance of the DPPA-NC-based PeLED reaches 23,480 cd m$^{-2}$ at a current density of ~500 mA cm$^{-2}$, while that of the pristine-NC-based PeLED only achieves 5710 cd m$^{-2}$. Correspondingly, the DPPA-NC-based PeLED reaches a peak EQE of 24.8%, much higher than the control device with a peak EQE of 12.1% (Fig. 3c). We note that the introduced triphenylphosphonium iodide (TMPI) in the PeLED further passivates the NC films, slightly improving the optical properties and enhancing the device performance (Supplementary Figs. 7, 14). The histograms of 42 devices based on the DPPA-NCs show a mean peak EQE of 22.8% with a standard deviation of 0.7% and an average maximum luminance of 23,690 cd m$^{-2}$ (Fig. 3d). To the best of our knowledge, the brightness of the DPPA-NC-based device sets a record among the PeLEDs with red emission (Fig. 3e and Supplementary Table 4). The measured half-lifetime ($T_{50}$, the time reaches 50% of the initial brightness) at an initial luminance of 100 cd m$^{-2}$ for the device based on the DPPA-NCs is around 13.1 h (Fig. 3f), 17 times longer than that of the PeLEDs based on the pristine-NCs (around 0.76 h).

We then measured the EL spectra of the two devices at various current densities to assess their spectral stability. As illustrated in Fig. 3g, the EL spectra of the control device exhibit a redshift, accompanied by notable emission broadening, as the operating current densities increase. This phenomenon is attributed to the severe halide ions migration, particularly the different ion migration rates between the Br and I ions under an electric field. In contrast, the EL spectra of the DPPA-NC-based PeLED, which exhibits increased ion migration activation energy (Supplementary Fig. 15 and Supplementary Table 5), maintain a nearly identical shape with a negligible peak shift and slight broadening within 1000 mA cm$^{-2}$ (Fig. 3h). More importantly, this stability persists even at a high current density of 100 mA cm$^{-2}$ for a long time, whereas the EL peak of the device based on pristine-NCs exhibits a noticeable redshift within 2 min under similar conditions (Fig. 3i, j). Temperature is a crucial factor influencing ion migration behavior. An increase in temperature usually accelerates ion migration, compromising device performance. The improved luminance and EL stability motivate us to monitor the surface temperature variation of the device at a fixed current density of 1000 mA cm$^{-2}$ using an infrared-thermal-imaging-camera. As depicted in Fig. 3k, l, the DPPA-NC-based PeLED exhibits a temperature of 57.8 °C after operation at 1000 mA cm$^{-2}$ for 30 s, while the control devices experience a rapid temperature increase to 78.2 °C. This result evidences the reduced Joule heat generation and enhanced thermal dissipation properties of the DPPA-NC films, laying the foundation for achieving high-brightness devices at high current densities. Furthermore, owing to the improved carrier transport of the DPPA-NC films, the devices can be operated at high current density and exhibit reduced efficiency roll-off. The EQE drops from 24.8 to 14% as the current density and brightness increase to 101 mA cm$^{-2}$ and 10,633 cd m$^{-2}$ (Table 1).

## PeLEDs on sapphire substrate and operated in pulsed mode

The superior EL performance demonstrated by the DPPA-NCs suggests its potential as the ideal candidate for efficient and ultrabright pure red PeLEDs. A slight broadening is still observed in the short-wavelength region of the EL spectra. This phenomenon is attributed to

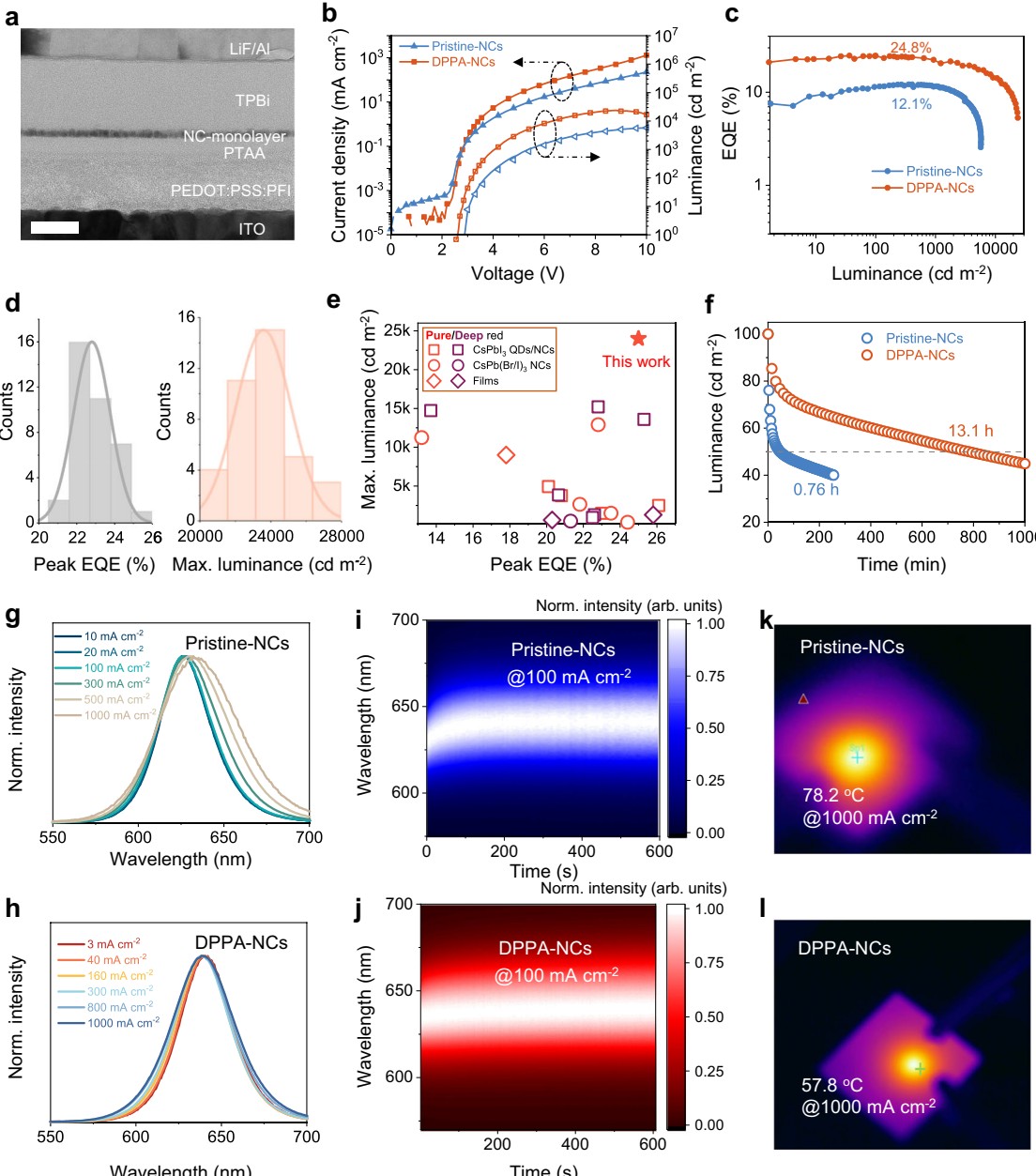

**Fig. 3 | Performance of the CsPb(Br/I)₃ NC-based LEDs. a** Cross-sectional TEM image of the multi-layered DPPA-NC-based LED. Scale bar: 50 nm. **b** Current density-voltage-luminance curves for the PeLEDs. **c** EQE−luminance characteristics for the PeLEDs. **d** Statistical distribution histograms of peak EQE and maximum luminance summarized from 42 DPPA-NC-based LED devices. **e** Reported peak EQE and maximum luminance of red-emissive (pure red and deep red) LEDs. The star stands for this work. **f** The operational $T_{50}$ of the PeLEDs at an initial luminance of 100 cd m⁻². **g, h** Normalized EL spectra of the PeLEDs under different current densities based on the pristine-NCs and DPPA-NCs, respectively. **i, j** EL spectra mapping of PeLEDs at 100 mA cm⁻². **k, l** Infrared-thermal-imaging pictures of the PeLEDs after operating at 1000 mA cm⁻² for 30 s.

temperature-dependent broadening, originating from the inevitable generation of Joule heat and insufficient heat dissipation of the device. The real-time PL spectra of the DPPA-NC films with increasing temperature evidence a similar short-wavelength spectral broadening (Fig. 4a). This observation implies that the performance might be further improved through effective thermal management of the device, specifically by accelerating the dissipation of Joule heat generated during device operation. To address this issue, we opted for a transparent sapphire substrate ($\lambda = 46$ W m⁻¹K⁻¹) with high-thermal conductivity instead of glass ($\lambda = 1.1$ W m⁻¹K⁻¹) for PeLEDs. To assess the effectiveness of sapphire as a heat sink for PeLEDs, we monitored the surface temperature of the device using an infrared-thermal camera under the same operating conditions as above. Remarkably, the

surface temperature of the device decreases from 57.8 to 41.4 °C by changing the substrate (Fig. 4b), indicating efficient dissipation of Joule heat throughout the entire substrate (Supplementary Fig. 16). Consequently, the sapphire-based PeLEDs exhibit excellent spectral stability (Fig. 4c). The brightness of the sapphire-based PeLEDs device presents a sharp rise alongside the increase of current density, as depicted in Fig. 4d. At lower current densities (<200 mA cm⁻²), the luminance−current density characteristics of sapphire- and glass-based devices almost coincide. However, discernible distinctions arise at higher current densities. The luminance of the LED based on the sapphire substrate continues growing until the current density reaches 1200 mA cm⁻² with a maximum brightness of 35,120 cd m⁻². In contrast, the LED based on the glass substrate manifests a declining trend at

**Table 1 | Summary of EQEs under various brightness for different PeLEDs in this work**

| PeLEDs | EQE$_{100\,cd\,m^{-2}}$ | EQE$_{1\,k\,cd\,m^{-2}}$ | EQE$_{5\,k\,cd\,m^{-2}}$ | EQE$_{10\,k\,cd\,m^{-2}}$ | EQE$_{50\,k\,cd\,m^{-2}}$ |
|---|---|---|---|---|---|
| **Pristine**[a] | 11.45% | 11.6% | 5.3% | / | / |
| **DPPA**[b] | 24.7% | 22.1% | 17.8% | 14.4% | / |
| **Sapphire**[c] | 24.5% | 24.2% | 20.4% | 17.2% | / |
| **Pulsed mode**[d] | / | / | / | 18% | 23.4% |

[a]The LED based on the pristine-NCs on glass operated in D.C. mode.
[b]The LED based on the DPPA-NCs on glass operated in D.C. mode.
[c]The LED based on the DPPA-NCs on sapphire operated in D.C. mode.
[d]The LED based on the DPPA-NCs on sapphire operated in pulsed mode.

approximately 500 mA cm$^{-2}$. This underscores the significance of the superior thermal conductivity of the device in enhancing the overall performance of LEDs, particularly under high current densities. Furthermore, since the peak EQE is attained at lower current densities (~1 mA cm$^{-2}$), there is no evident discrepancy in the maximum efficiency between the two PeLEDs (Supplementary Figs. 17, 18).

Although devices optimized with the DPPA-NCs emitting layer on glass substrates demonstrate a commendable EQE roll-off, the efficiency droop further improves when transferring the devices on sapphire substrates. The explicit comparison of efficiency roll-off could be derived from the curves of normalized EQE versus luminance, as depicted in Fig. 4e. Quantitatively, it sustains 72% of the peak efficiency at a brightness of 10,000 cd m$^{-2}$. Remarkably, even at a brightness of

20,000 cd m$^{-2}$, it maintains 55% of the peak efficiency, corresponding to an absolute EQE of over 10% (Table 1). We further introduce a more critical parameter, $L_{EQE-90\%}$[43,44], that is, the luminance at which the EQE has dropped to 90% of its maximum value, to compare the EQE roll-off of the PeLEDs. Typically, the $L_{EQE-90\%}$ lies in 2300 cd m$^{-2}$ for the sapphire-based LED, 4.6 times higher than that on the glass, and surpasses most pure red PeLEDs reported in the literature, representing the state-of-the-art devices (Supplementary Table 4). Besides, by replacing the glass substrate with sapphire, the $T_{50}$ is also increased to around 20 h (Fig. 4f).

Pulsed voltage measurements provide another approach to relieving Joule heating, affording adequate time for effective heat dissipation[45]. Figures 4g–i show the characteristics of PeLEDs driven in a pulse width of 2 ms with a duty cycle of 10% (pulse frequency: 50 Hz). As illustrated in Fig. 4g and Supplementary Fig. 19, it exhibits ultra-stable EL spectra within 15 A cm$^{-2}$. Figure 4h, i show the luminance–current density and EQE-luminance curves of the pure red PeLEDs, respectively. Remarkably, a maximum luminance of approximately 390,000 cd m$^{-2}$ at 7.8 A cm$^{-2}$, and a $L_{EQE-90\%}$ of 70,640 cd m$^{-2}$ is achieved for the pure red PeLEDs, two orders of magnitude brighter than the previous reports (Supplementary Table 4). Furthermore, even with a larger pulse duty of 50%, it exhibits enhanced luminance of ~100,000 cd m$^{-2}$ and suppressed EQE roll-off compared to D.C. mode (Supplementary Fig. 20). The inset image in Fig. 4h depicts the bright EL of the pulse-driven PeLED under 1.0 A cm$^{-2}$, as bright as an inorganic

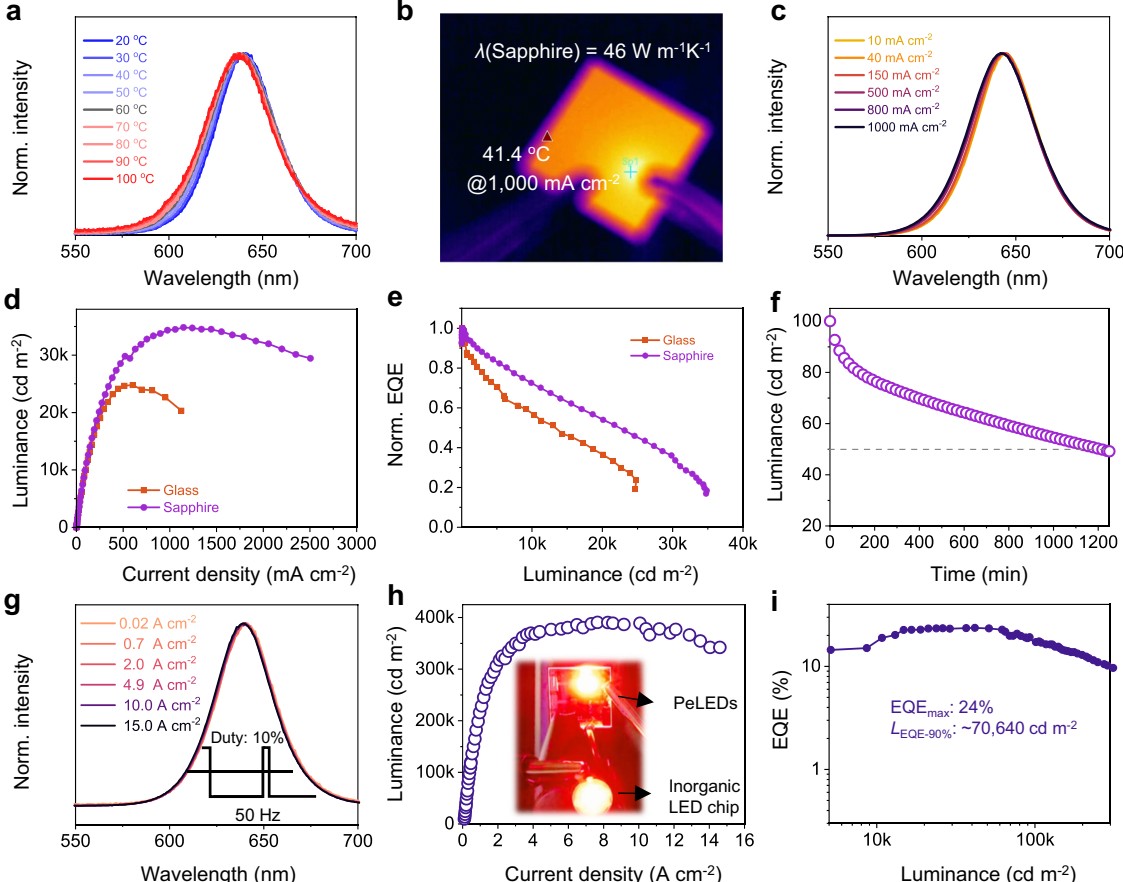

**Fig. 4 | Performance of PeLEDs based on sapphire substrates and operated at pulsed mode. a** Normalized PL spectra of the DPPA-NC film with increasing temperatures. **b** Surface temperature images of the DPPA-NC-based PeLED on the sapphire substrate after operating at 1000 mA cm$^{-2}$ for 30 s. **c** Normalized EL spectra of PeLEDs on the sapphire substrate under different current densities. **d**, **e** Typical luminance–current density (**d**) and normalized EQE-luminance curves (**e**) of the PeLEDs based on glass and sapphire substrate. **f** $T_{50}$ of the PeLEDs on a

sapphire substrate. **g–i** Pulse mode operation of PeLED on sapphire substrate at a repetition rate of 50 Hz with a duty cycle of 10%. **g** Normalized EL spectra under different current densities. **h** Luminance–current density curves. The luminance refers to the brightness measured when the device is turned on. The inset in **h** shows a bright PeLED working at 1 A cm$^{-2}$ and an inorganic LED chip working at 1.42 A cm$^{-2}$ under pulsed operation. **i** EQE-luminance curve.

LED working at 1.42 A cm$^{-2}$. All these results illustrate that the excellent heat sink coupled with sufficient heat dissipation within devices can significantly enhance the brightness of PeLED, and effectively address EL spectra instability and efficiency roll-off.

In summary, this work emphasizes the significance of thermal management in advancing high-performance PeLEDs. Synergistic strategies by manipulating Joule heat generation in NC emissive layer and enhancing thermal dissipation in devices were developed, yielding an ultra-high brightness of 35,120 cd m$^{-2}$, a peak EQE of ~25%, an operation lifetime of ~20 h, and suppressed EQE roll-off, representing the state-of-the-art pure red PeLEDs. Comprehensive analyses indicate that the enhanced optical-electro properties contribute to reducing Joule heat generation, and effective heat sink integration alongside ample dissipation facilitates efficient thermal management within the device. Future endeavors could prioritize elucidating mechanisms to minimize Joule heating generation and enhance heat dissipation capabilities, such as improving the thermal conductivity of charge-transport layers, or nanopatterning the current injection area, providing a universal strategy for advancing applicable PeLEDs.

## Methods

### Materials
Cesium carbonate (Cs$_2$CO$_3$, 99.95%), and lead (II) iodide (PbI$_2$, 99.999%) were purchased from Sigma Aldrich. Zinc iodide (ZnI$_2$, 99.99%), Zinc bromide (ZnBr$_2$, 99.9%), Diphenyl azidophosphate (DPPA, 97%), diphenyl phosphate (DPP, ≥98%), oleylamine (OAm, 80–90%), methyl aceate (99%), methyl triphenylphosphonium iodide (TMPI, 98%), nafion perfluorinated resin solution (PFI, 5 wt.% in a mixture of lower aliphatic alcohols and water, contains 45% water), and lithium fluoride (LiF, 99.99%) were purchased from Shanghai Macklin Biochemical Co., Ltd. m-Xylenes (99%) and n-octane (99%, Superdry) were purchased from J&K Scientific. 1-octadecene (ODE, 90%) was purchased from Acros Organics and oleic acid (OA, 90%) was purchased from Alfa Aesar. Poly(2,3-dihydrothieno-1,4-dioxin)-poly(-styrene sulfonate) (PEDOT: PSS), poly[bis(4-phenyl) (2,4,6-trimethylphenyl) amine] (PTAA, Mn >25,000) and 1,3,5-tris(1-phenyl-1H-benzimidazol-2-yl) benzene (TPBi, >99%) were purchased from Xi'an Polymer Light Technology Corp. All chemicals were used directly without any further purification.

### Preparation of cesium precursor
The Cs-oleate precursor was prepared following the modified literature-reported method[27,46]. Typically, Cs$_2$CO$_3$ (0.307 mmol, 100 mg), OA (0.4 mL), and ODE (3.5 mL) were loaded into a 25 mL three-necked flask. The mixture was degassed for half an hour at 80 °C, then heated to 120 °C for another 30 min under a continuous nitrogen flow.

### Synthesis of CsPb(Br/I)$_3$ nanocrystals
For the synthesis of DPPA-NCs, PbI$_2$ (0.19 mmol, 88 mg), ZnI$_2$ (1.0 mmol, 319 mg), ZnBr$_2$ (0.742 mmol, 167 mg), DPPA (0.3 mL), OA (2 mL), OAm (2.4 mL), and ODE (5 mL) were added to a 25 mL three-necked flask. The mixture underwent degassing at room temperature for 30 min before being heated to 170 °C for an additional 10 min under a constant nitrogen flow. Subsequently, 0.4 mL of the Cs-oleate precursor was quickly injected into the flask. Five seconds after this injection, the reaction mixture was rapidly cooled by immersing the flask in ice water. The synthesis of the pristine nanocrystals (NCs) followed the same procedure as that of the DPPA-NCs, but without the inclusion of DPPA.

### Purification of CsPb(Br/I)$_3$ nanocrystals
To the crude solution, 5 mL of m-xylenes and 10 mL of methyl acetate were added, followed by centrifugation at 8600 × g for 3 min to remove unreacted products, byproducts, and other impurities. The precipitate was discarded, and 20 mL of methyl acetate was added to the supernatant. After another round of centrifugation at 8600 × g for 3 min, the resulting precipitate was re-dispersed in 1 mL of ultradry xylenes. The dispersed nanocrystals were then washed with 20 mL of methyl acetate and centrifuged again for 3 min at 8600 × g. Finally, the precipitate was re-dispersed in n-octane and filtered through a 0.22 μm PTFE filter. The final NC solution was stored for further characterization and LED fabrication. For $^1$H NMR characterizations, the final NCs were dispersed in CDCl$_3$.

### Device fabrication
The patterned ITO glass substrates were sequentially cleaned with acetone, deionized water, and ethanol using ultrasonication for 15 min each. Following a 20-min surface treatment with oxygen plasma, a PEDOT:PSS solution premixed with Nafion (1:1 volume ratio) was spin-coated onto the substrate, first at 500 rpm for 5 s, then at 4000 rpm for 40 s. This was followed by annealing at 160 °C for 15 min under ambient conditions. Subsequently, a PTAA solution in chlorobenzene (5 mg mL$^{-1}$) was spin-coated onto the PED-OT:PSS:PFI layer at 2000 rpm for 60 s and then annealed at 170 °C for 20 min under a nitrogen atmosphere. Then the purified NCs in octane (5 mg mL$^{-1}$) mixed with TMPI-toluene saturated solution (volume ratio 10:1) were spin-coated at 4000 rpm for 60 s to achieve monolayer NC film and followed by annealing at 60 °C for 5 min. The key to preparing monolayer thin films is to control the concentration of NC solution and the speed of spin coating. The TMPI molecules further passivate the surface of the NCs and slightly improve the NC film properties and device performance (Supplementary Figs. 7, 14). Finally, 80 nm of the TPBi layer, 1 nm of the LiF layer, and 100 nm of the Al electrode were deposited by a thermal evaporation system equipped with a shadow mask under a high vacuum. An overlapping area of the ITO and Al electrodes of 1.94 mm$^2$ was used to define the working area of the device. The fabrication of the sapphire-based PeLEDs followed the same process, except patterned ITO sapphire substrates were utilized.

### Device characterizations
The current density-luminance-voltage characterizations were measured by a Keithley 2400 electrometer and an integration sphere coupled with a QE Pro spectrometer[47]. The $T_{50}$ was measured using a commercialized aging system (Guangzhou New Vision Optoelectronic Technology Co. Ltd.).

### PeLEDs measurement in pulsed voltage mode
Square voltage pulses with 2 ms or 10 ms width at a repetition rate of 50 Hz were generated by the Keithley 2635B and applied to the device. Resistors of 1, 10, or 100 Ω were placed in series with the device. Transient voltages were measured using a dual-channel oscilloscope (TDS 2024 C, Tektronix) and the transient current was calculated from the voltage across the resistor in series with the device. A Si avalanche photodetector (APD 130A2/M, Thorlabs) was utilized to collect the electroluminescence response of the device. The actual brightness and EQE were corrected by the D.C. response using that from the standard LED characterization setup. For pulsed mode operation, the luminance refers to the brightness measured when the device is turned on.

The conductivity measurement was carried out on the lateral devices with a structure of glass/QDs/Au. The CsPb(Br/I)$_3$ NCs (~15 mg mL$^{-1}$ in octane) were spin-coated onto the glass at 4000 rpm for 45 s. Then 80 nm of Au was deposited using a mask with an interdigital electrode pattern by thermal evaporation under a high vacuum (<5 × 10$^{-4}$ Pa). The channel width and the stripe length pattern are 150 μm and 7.4 mm, respectively. Keithley 2635B was used to detect the signal and current voltage.

## Data availability

The data that support the findings of this study are provided in the Supplementary Information/Source Data file. Source data are provided with this paper.

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

## Acknowledgements

This work was financially supported by the Fundamental Research Funds for the Central Universities (K20240042, X.D.), the National Key Research and Development program (2023YFB3608902, X.D.), the "Pioneer" and "Leading Goose" R&D Program of Zhejiang (2024C01191, X.D.), the National Natural Science Foundation of China (52102188, X.D.; U22A20133, Z.Y.), the Shanxi-Zheda Institute of Advanced Materials and Chemical Engineering (2022SZ-TD004, H.H.), Wenzhou key scientific and technological innovation research projects (ZG2023039, Z.Y.), and Science and Technology Projects of the Institute of Wenzhou, Zhejiang University (XMGL-KJZX-202302, X.D.). X. Dai gratefully acknowledges the support of the Zhejiang University Education Foundation Qizhen Scholar Foundation. We thank Mr. Yifan He for his assistance in device fabrication.

## Author contributions

X.D. and H.L. conceived the idea. H.L. designed the experiments, synthesized the nanocrystals, and fabricated the device, characterization, and data analysis. X.Z. synthesized the nanocrystals, fabricated the device, and assisted in characterization. D.Z. and Y.G. assisted in device characterization. Y.F. assisted in the synthesis of nanocrystals. Z.M. assisted in optical characterization. J.H. and H.H. provided helpful discussion. H.L. wrote the first draft of the manuscript. X.D. and Z.Y. revised the manuscript. X.D. and Z.Y. supervised the study. All the authors discussed the results and commented on the manuscript.

## Competing interests

The authors declare no competing interests.
