## [Peer Review File · Nature Communications]

Thermal Management Towards Ultra-bright and Stable
Perovskite Nanocrystal-based Pure Red Light-Emitting DiodesREVIEWER COMMENTS

Reviewer #1 (Remarks to the Author):

In this manuscript, the authors reported multiple thermal management strategies to improve the brightness and lifetime of pure red perovskite LEDs. While the use of sapphire substrates and pulsed voltage mode is not new, using DPPA to regulate the anchoring ligands of CsPb(Br/I)₃ NCs during synthesis which led to enhanced optical properties and carrier transport performance is novel. The authors did substantial material characterizations, also performed PeLED testing which showed very high brightness, peak EQE, reduced EQE roll-off and increased operational lifetime. There are a few unclear points in the manuscript. It is recommended that the authors address them to enhance the manuscript.

1. Page 10: “the EL spectra of the control device exhibit a redshift, accompanied by notable emission broadening, as the operating current densities increase. In contrast, the EL spectra of the DPPA-NCs-based PeLED maintain a nearly identical shape with a negligible peak shift and slight broadening within 1000 mA cm⁻².” Please explain what caused the difference.
2. For pulsed mode operation, is the reported luminance average luminance or calculated for only the period when the device is turned on?
3. The inset image in Fig. 4h of PeLED EL under 1 A cm⁻², is this under DC or pulsed operation?

Reviewer #2 (Remarks to the Author):

Perovskite nanocrystals (NCs) are recognized as promising materials for next-generation light-emitting applications; however, achieving high performance in pure red emission remains an ongoing challenge. The manuscript reports on the achievement of ultra-bright and stable pure red LEDs through the utilization of diphenylphosphoryl azide-mediated regulation of the NC surface to reduce Joule heat generation, coupled with highly thermally conductive sapphire substrates to enhance thermal dissipation in devices. As a result, the authors have developed an efficient and stable pure red LED with a record-high brightness of 35 120 cd m⁻², a peak EQE of ~25%, an operation lifetime of ~20 hours, and suppressed EQE roll-off. These results are highly impressive and represent the state-of-the-art in pure red PeLEDs. The characterization of the materials and devices is comprehensive, and the performance of the devices is significant. These findings are not only encouraging but also suggest the potential for further advancements in PeLEDs using a similar strategy. I highly recommend the manuscript for publication in Nature Communications. I would suggest the authors consider the following questions during their revision.

- 1) As mentioned in the paper that the accumulated heat can be alleviated by minimizing nonradiative recombination losses, it is encouraged that the authors provide the nonradiative recombination rates for both DPPA-NCs and pristine-NCs.
- 2) The authors mentioned that the PeLEDs were constructed with monolayer NC film. It is encouraged that the authors elaborate on the method used to achieve the monolayer NC films and provide evidence supporting the formation of monolayers.

- 3)The authors should provide a description of the inset photograph in Fig. 4h.
- 4)The authors are encouraged to provide an explanation for why the electroluminescence spectra of the control device exhibit a significant redshift and broadening.
- 5)The authors are encouraged to cite a recent paper on perovskite LEDs published in Nature Photonics (doi.org/10.1038/s41566-024-01382-6).

Reviewer #3 (Remarks to the Author):

This study presents high brightness and stable pure red perovskite nanocrystal light-emitting diodes achieved through surface ligand engineering and sapphire substrates with high thermal conductivity. While the device performance is noteworthy, the novelty may not meet the standards for publication in Nature Communications. Furthermore, additional unresolved issues warrant consideration.

1. The EL spectra of DPPA-NCs-based PeLEDs exhibit minimal peak shift compared to the control device. What factors contribute to this enhancement in spectral stability? Is there a correlation between the consistent EL spectra and effective thermal management strategies?
2. The PLQY of the film significantly impacts the EQE of the device. While the authors reported a near-unity PLQY of DPPA-NCs in solution, what is the PLQY of the perovskite nanocrystal film?
3. The reviewer noted that the authors fabricated the perovskite film with TMPI. What are the implications of TMPI on both the film properties and device performance?
4. Thermal stability plays a crucial role in both efficiency roll-off and operational stability of the device. What is the thermal stability comparison between pristine perovskite nanocrystals and those treated with DPPA? Providing supporting evidence would strengthen this assessment.
5. Considering that thermal management contributes to high brightness and stability, one might expect an improvement in EQE. However, if high thermal conductivity of sapphire substrates helps mitigate efficiency roll-off, it's puzzling that the EQE of sapphire-based devices is lower than that of glass-based ones. What factors might explain this discrepancy?
6. The authors did not do a good job of background checks. To date, tens of reports about thermal management strategies have been employed in perovskite nanocrystal LEDs. The reviewer strongly recommends the authors should summarize previous reports in the Introduction Section to explain the advance of this work.

Responses to Reviewer #1

In this manuscript, the authors reported multiple thermal management strategies to improve the brightness and lifetime of pure red perovskite LEDs. While the use of sapphire substrates and pulsed voltage mode is not new, using DPPA to regulate the anchoring ligands of CsPb(Br/I)₃ NCs during synthesis which led to enhanced optical properties and carrier transport performance is novel. The authors did substantial material characterizations, also performed PeLED testing which showed very high brightness, peak EQE, reduced EQE roll-off and increased operational lifetime. There are a few unclear points in the manuscript. It is recommended that the authors address them to enhance the manuscript.

Q1-1. Page 10: “the EL spectra of the control device exhibit a redshift, accompanied by notable emission broadening, as the operating current densities increase. In contrast, the EL spectra of the DPPA-NCs-based PeLED maintain a nearly identical shape with a negligible peak shift and slight broadening within 1000 mA cm⁻².” Please explain what caused the difference.

Our revision and responses: We appreciate the reviewer for his/her positive comments on the scientific quality of our manuscript. **The broadening and redshift observed in the EL spectra of the control device are attributed to the migration of halide ions under an electric field.** In perovskite materials, particularly those using a mixed Br/I system, the ion migration rates of the different halides are not uniform. This disparity leads to halide segregation within the film, which in turn causes changes in the emission spectra. This phenomenon has been well-documented in the literature (Chem. Mater., 2017, 29(14): 5965-5973; J. Am. Chem. Soc., 2019, 141(3):1269-1279).

For the DPPA-NC-based PeLEDs, the near-constant shape and negligible peak shift in the EL spectra, even at high current densities up to 1000 mA cm⁻², can be attributed to the surface modification of the nanocrystals, which effectively suppresses ion migration. Our investigations of ion migration activation energy confirmed that the DPPA-NCs exhibit an increased activation energy of 212 meV, higher than the 145 meV for the pristine-NC film (Fig. R1-1, Table R1-1). Besides, the DPPA-NC-based PeLEDs show a lower working temperature than the control devices (Figs. 3k and 3l), which arises from the reduced Joule heat generation.

Temperature is a crucial factor influencing ion migration behaviour. An increase in temperature usually accelerates ion migration, compromising device performance.

We have included the ion migration activation energy measurements in our revised manuscript and added discussions (Page 11), which reads, “As illustrated in Fig. 3g, the EL spectra of the control device exhibit a redshift accompanied by notable emission broadening, as the operating current densities increase. This phenomenon is attributed to the severe halide ions migration, particularly the different ion migration rates between the Br and I ions under an electric field. In contrast, the EL spectra of the DPPA-NC-based PeLED, which exhibits increased ion migration activation energy (Supplementary Fig. 15 and Supplementary Table 5), maintain a nearly identical shape with a negligible peak shift and slight broadening within 1000 mA cm⁻² (Fig. 3h). More importantly, this stability persists even at a high current density of 100 mA cm⁻² for a long time, whereas the EL peak of the device based on pristine-NCs exhibits a noticeable redshift within 2 minutes under similar conditions (Figs 3i and 3j). Temperature is a crucial factor influencing ion migration behaviour. An increase in temperature usually accelerates ion migration, compromising device performance”.

We have also added one sentence in the Abstract, “These merits inhibit ion migration of the CsPb(Br/I)₃ NC film, promising outstanding spectra stability”.

Fig. R1-1 (Supplementary Fig. 15) | Ion migration activation energy measurement. a–b, The time-dependent currents from 255 K to 305 K by applying bias at 12 V for the (a) pristine-NC film, and (b) DPPA-NC film, respectively. **c–d,** Temperature-dependent plots of $\ln(kT)$ versus $1,000/T$ for the (c) pristine-NC film, and (d) DPPA-NC film, respectively.

Table R1-1 (Supplementary Table 5) | The fitting data from the current decay plots of the pristine-NC film and DPPA-NC film.

Temperature	Pristine-NCs		DPPA-NCs	
	τ_1 (s)	τ_2 (s)	τ_1 (s)	τ_2 (s)
255 K	0.71	45.20	0.73	49.09
260 K	0.71	40.24	0.61	34.95
265 K	0.76	37.43	0.67	29.82
270 K	0.80	33.06	0.75	28.51
275 K	0.83	32.66	0.77	24.34
280 K	0.87	26.8	0.75	20.54
285 K	0.92	26.22	0.81	19.29
290 K	0.95	24.89	0.79	16.78
295 K	0.94	20.62	0.76	13.52
300 K	0.93	21.15	0.75	12.09
305 K	0.86	17.84	0.72	11.43

The method of ion migration activation energy measurement was added in the experimental section (Supplementary Information, page 27):

Ion migration activation energy measurement

The decay in the temperature-dependent temporal response can reflect the kinetics of ionic movement. The temperature-dependent ionic conductivity σ was addressed as²⁰: $\sigma(T) = \sigma_0 \exp(-\frac{E_a}{k_B T})$. Where E_a is the activation energy for ion transport, k_B is the Boltzmann constant.

The decay rate ($k = \tau^{-1}$) of the current decay represents the ionic transport dynamics and is proportional to the ionic conductivity. Therefore, the temperature-dependent decay rates can be used to obtain the thermal active energy for ion migration.

The ion migration activation energy measurement was carried out on the device with a structure of glass/NCs/Au. The NC solution ($\sim 15 \text{ mg mL}^{-1}$ in octane) was spun onto the glass at 4000 rpm for 45s. Then the Au interdigital electrode (80 nm) was deposited onto the NC film by thermal evaporation under a high vacuum ($< 5 \times 10^{-4}$ Pa). The samples were placed on a liquid nitrogen thermostat, and Keithley 2635B was used to detect the current signal. The current decay curves were obtained at 12 V and fitted by a double exponential function. The τ_1 is independent of temperature, which relates to the equipment response. The τ_2 represents the time constant of ion migration and is used for calculating the activation energy. The ion migration activation E_a was calculated as the slope of $\ln(kT)-1/T$ using the relation $\ln(kT) = C - \frac{E_a}{k_B T}$, where k can be obtained by $k = \tau^{-1}$ using the time constant τ_2 at different temperatures from 255 K to 305 K.

Q1-2. For pulsed mode operation, is the reported luminance average luminance or calculated for only the period when the device is turned on?

Our revision and responses: In pulsed mode operation, the reported luminance refers to the brightness measured during the period when the device is turned on.

For the PeLEDs measurement in pulsed voltage mode, transient photovoltages of the electroluminescence were collected by a Si photodetector and measured using a dual-channel oscilloscope. The actual luminance was corrected by the D.C. response using that from the standard LED characterization setup. **To clearly describe our test, we added a sentence on Page 17, which reads** “The actual brightness and EQE were corrected by the D.C. response using that from the standard LED characterization setup. **For pulsed mode operation, the luminance refers to the brightness measured when the device is turned on.**”. **We also revised the caption of Fig. 4h, which reads** “**h, Luminance-current density curves. The luminance refers to the brightness measured when the device is turned on. The inset in h shows a bright PeLED working at 1 A cm^{-2} and an inorganic LED chip working at 1.42 A cm^{-2} under pulsed operation**”.

Q1-3. The inset image in Fig. 4h of PeLED EL under 1 A cm^{-2} , is this under DC or pulsed operation?

Our revision and responses: The inset image of the working PeLED is under pulsed operation. We also clarified this point on page 14, which reads “The inset image in Fig. 4h depicts the bright EL of the pulse-driven PeLED under 1.0 A cm^{-2} , as bright as an inorganic LED working at 1.42 A cm^{-2} .”.

Responses to Reviewer #2

Perovskite nanocrystals (NCs) are recognized as promising materials for next-generation light-emitting applications; however, achieving high performance in pure red emission remains an ongoing challenge. The manuscript reports on the achievement of ultra-bright and stable pure red LEDs through the utilization of diphenylphosphoryl azide-mediated regulation of the NC surface to reduce Joule heat generation, coupled with highly thermally conductive sapphire substrates to enhance thermal dissipation in devices. As a result, the authors have developed an efficient and stable pure red LED with a record-high brightness of 35 120 cd m⁻², a peak EQE of ~25%, an operation lifetime of ~20 hours, and suppressed EQE roll-off. These results are highly impressive and represent the state-of-the-art in pure red PeLEDs. The characterization of the materials and devices is comprehensive, and the performance of the devices is significant. These findings are not only encouraging but also suggest the potential for further advancements in PeLEDs using a similar strategy. I highly recommend the manuscript for publication in Nature Communications. I would suggest the authors consider the following questions during their revision.

Q2-1. As mentioned in the paper that the accumulated heat can be alleviated by minimizing nonradiative recombination losses, it is encouraged that the authors provide the nonradiative recombination rates for both DPPA-NCs and pristine-NCs.

Our revision and responses: We appreciate the reviewer for his/her positive comments on the scientific quality of our manuscript. According to the reviewer's suggestion, we have calculated nonradiative recombination rates (K_{nr}) for both pristine-NCs and DPPA-NCs. As indicated in Tables R2-1 and R2-2, the DPPA-NCs solution and film show a slower K_{nr} than the pristine-NCs. **In the revised manuscript, we have included the corresponding results and discussion in Page 7, Paragraph 2, which reads, "In contrast, the TRPL of the pristine-NCs can only be fitted by bi-exponential decay along with the shortened effective lifetime of 11.8 ns and a much longer nonradiative recombination rates (K_{nr}) (Fig. 2b, Supplementary Fig. 7, Supplementary Table 1 and Table 2)".**

The calculation method of the K_{nr} was included in Supplementary information, Page 26, which reads "For the calculation of the nonradiative recombination rates, PL decays were

fitted by the biexponential function:

$$A(t) = A_1 \exp\left(-\frac{t}{\tau_1}\right) + A_2 \exp\left(-\frac{t}{\tau_2}\right) + A_0$$

where A_0 , A_1 , and A_2 are constants, t is time and τ_1 , τ_2 are the decay times. The average PL lifetime (τ_{av}) was calculated as:

$$\tau_{av} = \frac{A_1 \tau_1^2 + A_2 \tau_2^2}{A_1 \tau_1 + A_2 \tau_2} = f_1 \tau_1 + f_2 \tau_2$$

Where:

$$f_1 = \frac{A_1 \tau_1}{A_1 \tau_1 + A_2 \tau_2}$$

$$f_2 = \frac{A_2 \tau_2}{A_1 \tau_1 + A_2 \tau_2}$$

Photoluminescence quantum yield in low-dimensional perovskite under low excitation power density can be described by the following formula:

$$PLQY = \frac{k_r}{k_{nr} + k_r}$$

The carrier decay time and carrier recombination rate (k_r , k_{nr}) have the following relationship:

$$k_{nr} + k_r = \frac{1}{\tau}$$

Thus, the radiative recombination rate: $k_r = \frac{QY}{\tau_{av}}$, and the nonradiative recombination rate:

$$k_{nr} = \frac{1-QY}{\tau_{av}}$$

Ref. 19 Yang, S. et al. Electron Delocalization in CsPbI₃ Quantum Dots Enables Efficient Light-Emitting Diodes with Improved Efficiency Roll-Off. Adv. Opt. Mater., 2200189 (2022).

Table R2-1 (Supplementary Table 1) | Summary of time-resolved PL exponential fitting parameters for solutions of the pristine-NCs and DPPA-NCs.

Solutions	f ₁ (%)	τ ₁ (ns)	f ₂ (%)	τ ₂ (ns)	τ _{effective} (ns) ^a	τ _{av} (ns)	PLQY (%)	K _{nr} × 10 ⁷ (s ⁻¹)
Pristine-NCs	72.78	12.12	27.22	42.17	11.8	20.57	78%	1.06
DPPA-NCs	100	20.9			22.3	20.9	98%	0.096

^a The effective decay time ($\tau_{effective}$) is extracted from the raw data, representing the time when the peak intensity drops to 1/e.

Table R2-2 (Supplementary Table 2) | Summary of time-resolved PL exponential fitting parameters for films of the pristine-NCs and DPPA-NCs with or without TMPI.

Films	f_1 (%)	τ_1 (ns)	f_2 (%)	τ_2 (ns)	τ_{av} (ns)	PLQY (%)	$K_{nr} \times 10^7$ (s⁻¹)
Pristine-NCs-w/o TMPI	68.39	5.50	31.61	15.56	8.68	52%	5.53
Pristine-NCs+TMPI	70.16	6.44	29.8	20.71	10.69	58%	5.42
DPPA-NCs-w/o TMPI	98.53	10.87	1.47	59.75	11.6	87%	1.12
DPPA-NCs+TMPI	98.20	11.95	1.80	56.45	12.8	90%	0.78

Q2-2. The authors mentioned that the PeLEDs were constructed with monolayer NC film. It is encouraged that the authors elaborate on the method used to achieve the monolayer NC films and provide evidence supporting the formation of monolayers.

Our revision and responses: We have elaborated on the method used to achieve the monolayer NC film in the Methods section (Page 16, paragraph 2), which reads, “Then the purified NCs in octane (5 mg mL⁻¹) mixed with TMPI-toluene saturated solution (volume ratio 10:1) were spin-coated at 4000 rpm for 60 s to achieve monolayer NC film and followed by annealing at 60 °C for 5 min. The key to preparing monolayer thin films is to control the concentration of NC solution and the speed of spin coating.”.

The closely packed mono-emissive layer was evidenced by the cross-sectional TEM of the DPPA-NC-based LED, as displayed in Fig. R2-1. The NC layer demonstrated a thickness of 7.5 nm, corresponding to a single layer of the NC film. We have also included the TEM image in Fig. 3a. The method for obtaining the cross-sectional TEM images is detailed in our Supplemental information (Page 25, paragraph 1), which reads “Cross-sectional sample lamellae were cut and thinned down to electron transparency (~200 nm) with an FEI Helios Nanolab Dualbeam FIB/SEM following a standard protocol. The lamellae were then transferred directly into an FEI Osiris TEM operated at 200 kV.”

Fig. R2-1 (Fig. 3a) | Cross-sectional TEM image of the multi-layered DPPA-NC-based LED. Scale bar: 50 nm.

Q2-3. The authors should provide a description of the inset photograph in Fig. 4h.

Our revision and responses: We revised the sentence in our manuscript (Page 14, Paragraph 1), which reads, “The inset image in Fig. 4h depicts the bright EL of the pulse-driven PeLED under 1 A cm^{-2} , as bright as the inorganic LED working at 1.42 A cm^{-2} ”. We have also revised the caption of Fig. 4h, which reads “h, Luminance-current density curves. The luminance refers to the brightness measured when the device is turned on. The inset in h shows a bright PeLED working at 1 A cm^{-2} and an inorganic LED chip working at 1.42 A cm^{-2} under pulsed operation”.

Q2-4. The authors are encouraged to provide an explanation for why the electroluminescence spectra of the control device exhibit a significant redshift and broadening.

Our revision and responses: We appreciate the reviewer for his/her positive comments on the scientific quality of our manuscript. **The broadening and redshift observed in the EL spectra of the control device are attributed to the migration of halide ions under an electric field.** In perovskite materials, particularly those using a mixed Br/I system, the ion migration rates of

the different halides are not uniform. This disparity leads to halide segregation within the film, which in turn causes changes in the emission spectra. This phenomenon has been well-documented in the literature (Chem. Mater., 2017, 29(14): 5965-5973; J. Am. Chem. Soc., 2019, 141(3):1269-1279).

For the DPPA-NC-based PeLEDs, the near-constant shape and negligible peak shift in the EL spectra, even at high current densities up to 1000 mA cm^{-2} , can be attributed to the surface modification of the nanocrystals, which effectively suppresses ion migration. Our investigations of ion migration activation energy confirmed that the DPPA-NCs exhibit an increased activation energy of 212 meV, higher than the 145 meV for the pristine-NC film (Fig. R2-2, Table R2-3). Besides, the DPPA-NC-based PeLEDs show a lower working temperature than the control devices (Figs. 3k and 3l), which arises from the reduced Joule heat generation. Temperature is a crucial factor influencing ion migration behaviour. An increase in temperature usually accelerates ion migration, compromising device performance.

We have included the ion migration activation energy measurements in our revised manuscript and added discussions (Page 11), which reads “As illustrated in Fig. 3g, the EL spectra of the control device exhibit a redshift accompanied by notable emission broadening, as the operating current densities increase. This phenomenon is attributed to the severe halide ions migration, particularly the different ion migration rates between the Br and I ions under an electric field. In contrast, the EL spectra of the DPPA-NC-based PeLED, which exhibits increased ion migration activation energy (Supplementary Fig. 15 and Supplementary Table 5), maintain a nearly identical shape with a negligible peak shift and slight broadening within 1000 mA cm^{-2} (Fig. 3h). More importantly, this stability persists even at a high current density of 100 mA cm^{-2} for a long time, whereas the EL peak of the device based on pristine-NCs exhibits a noticeable redshift within 2 minutes under similar conditions (Figs. 3i and 3j). Temperature is a crucial factor influencing ion migration behaviour. An increase in temperature usually accelerates ion migration, compromising device performance”.

We have also added one sentence in the Abstract, which reads, “These merits inhibit ion migration of the $\text{CsPb}(\text{Br/I})_3$ NC film, promising outstanding spectra stability”.

Fig. R2-2 (Supplementary Fig. 15) | Ion migration activation energy measurement. a–b, The time-dependent currents from 255 K to 305 K by applying bias at 12 V for the (a) pristine-NC film, and (b) DPPA-NC film, respectively. **c–d,** Temperature-dependent plots of $\ln(kT)$ versus $1,000/T$ for the (c) pristine-NC film, and (d) DPPA-NC film, respectively.

Table R2-3 (Supplementary Table 5) | The fitting data from the current decay plots of the pristine-NC film and DPPA-NC film.

Temperature	Pristine-NCs		DPPA-NCs	
	τ_1 (s)	τ_2 (s)	τ_1 (s)	τ_2 (s)
255 K	0.71	45.20	0.73	49.09
260 K	0.71	40.24	0.61	34.95
265 K	0.76	37.43	0.67	29.82
270 K	0.80	33.06	0.75	28.51
275 K	0.83	32.66	0.77	24.34
280 K	0.87	26.8	0.75	20.54
285 K	0.92	26.22	0.81	19.29
290 K	0.95	24.89	0.79	16.78
295 K	0.94	20.62	0.76	13.52
300 K	0.93	21.15	0.75	12.09
305 K	0.86	17.84	0.72	11.43

The method of ion migration activation energy measurement was added in the experimental section (Supplementary Information, page 27):

Ion migration activation energy measurement

The decay in the temperature-dependent temporal response can reflect the kinetics of ionic movement. The temperature-dependent ionic conductivity σ was addressed as²⁰: $\sigma(T) = \sigma_0 \exp(-\frac{E_a}{k_B T})$. Where E_a is the activation energy for ion transport, k_B is the Boltzmann constant. The decay rate ($k = \tau^{-1}$) of the current decay represents the ionic transport dynamics and is proportional to the ionic conductivity. Therefore, the temperature-dependent decay rates can be used to obtain the thermal active energy for ion migration.

The ion migration activation energy measurement was carried out on the device with a structure of glass/NCs/Au. The NC solution (~15 mg mL⁻¹ in octane) was spun onto the glass at 4000 rpm for 45s. Then the Au interdigital electrode (80 nm) was deposited onto the NC film by thermal evaporation under a high vacuum ($< 5 \times 10^{-4}$ Pa). The samples were placed on a liquid nitrogen thermostat, and Keithley 2635B was used to detect the current signal. The current decay curves were obtained at 12 V and fitted by a double exponential function. The τ_1 is independent of temperature, which relates to the equipment response. The τ_2 represents the time constant of ion migration and is used for calculating the activation energy. The ion migration activation E_a was calculated as the slope of $\ln(kT)-1/T$ using the relation $\ln(kT) = C - \frac{E_a}{k_B T}$, where k can be obtained by $k = \tau^{-1}$ using the time constant τ_2 at different temperatures from 255 K to 305 K.

Q2-5. The authors are encouraged to cite a recent paper on perovskite LEDs published in *Nature Photonics* (doi.org/10.1038/s41566-024-01382-6).

Our revision and responses: Thanks for the reviewer's recommendation. **This article is beneficial for us to extend the strategy of stabilizing mixed halide perovskite. We have cited it in an appropriate place as Ref. 6.**

Ref 6. Yuan, S. *et al.* Efficient blue electroluminescence from reduced-dimensional perovskites. *Nat. Photonics* **18**, 425-431 (2024).

Responses to Reviewer #3

This study presents high brightness and stable pure red perovskite nanocrystal light-emitting diodes achieved through surface ligand engineering and sapphire substrates with high thermal conductivity. While the device performance is noteworthy, the novelty may not meet the standards for publication in Nature Communications. Furthermore, additional unresolved issues warrant consideration.

Our revision and responses: We appreciate the reviewer for his/her positive comments on the device's performance in this work and for pointing out helpful suggestions.

Regarding novelty, our research distinguishes it from conventional methods by employing a synergistic strategy that integrates surface ligand engineering of CsPb(Br/I)₃ nanocrystals with advanced thermal management. As the reviewer noted in Q3-6, numerous reports on thermal management strategies have been employed in PeLEDs, and we summarize these reports in the Introduction Section as suggested. **Previous studies have focused on enhancing heat dissipation through high-thermal-conductivity substrates or doping charge-transport layers on green-emissive PeLEDs. Here, the novelty of our work lies in the innovative use of diphenylphosphoryl azide (DPPA) for surface regulation of CsPb(Br/I)₃ nanocrystals.**

Firstly, this modification not only enhances the optical properties, significantly reducing non-radiative recombination rates, thereby effectively mitigating heat generation, but also improves carrier transport properties, lowering film resistance and alleviating Joule heat generation within the device. Secondly, achieving high-performance pure red emitter-based PeLEDs remains an unprecedented challenge due to their more complex preparation than their green counterparts. That is, mixed halide (Br/I) perovskite LEDs face issues with spectral instability caused by halide migration. The DPPA-mediate surface regulation of CsPb(Br/I)₃ nanocrystals effectively increases the energy barrier for ion migration, and the reduced Joule heat generation lowers the working temperature of the PeLEDs, which is beneficial for the spectra stability. Combined with the use of a high-thermal-conductivity sapphire substrate and the implementation of pulse driving mode, the thermal dissipation of the devices further improved.

This work underscores the critical importance of integrating surface regulation techniques with thermal management strategies to advance the performance of pure red PeLEDs, making a significant contribution to the field. Thus, we believe that the novelty of our revised manuscript meets the standards for publication in Nature Communications.

Besides the Introduction part, we also reorganized the Abstract to illustrate the novelty of this work more clearly.

Abstract:

Despite the promising candidacy of perovskite nanocrystals (NCs) for light-emitting diodes (LEDs), their pure red electroluminescence is hindered by low saturated luminance, severe external quantum efficiency (EQE) roll-off, and inferior operational stability. Here, we reported ultra-bright and stable pure red LEDs by manipulating Joule heat generation in the NC emissive layer and thermal management within the device. Diphenylphosphoryl azide-mediated regulation of the NC surface synergistically enhances the optical properties and carrier transport of the emissive layer, enabling reduced Joule heat generation and thus lowering the working temperature. These merits inhibit ion migration of the CsPb(Br/I)₃ NC film, promising outstanding spectra stability. Combined with the highly thermal-conductive sapphire substrates and implementation of pulse-driving mode, the pure red LEDs exhibit an ultra-bright luminance of 390 000 cd m⁻², a peak EQE of 25%, suppressed EQE roll-off, an operational half-life of 20 hours, and superior spectral stability within 15 A cm⁻².

Q3-1. The EL spectra of DPPA-NCs-based PeLEDs exhibit minimal peak shift compared to the control device. What factors contribute to this enhancement in spectral stability? Is there a correlation between the consistent EL spectra and effective thermal management strategies?

Our revision and responses: The enhanced spectral stability mainly originates from the increased ion migration activation energy of the NC film and the reduced generation of Joule heat in PeLED.

Firstly, the broadening and redshift observed in the EL spectra of the control device are attributed to the migration of halide ions under an electric field. In perovskite materials,

particularly those using a mixed Br/I system, the ion migration rates of the different halides are not uniform. This disparity leads to halide segregation within the film, which in turn causes changes in the emission spectra. This phenomenon has been well-documented in the literature (Chem. Mater., 2017, 29(14): 5965-5973; J. Am. Chem. Soc., 2019, 141(3):1269-1279).

For the DPPA-NC-based PeLEDs, the near-constant shape and negligible peak shift in the EL spectra, even at high current densities up to 1000 mA cm^{-2} , can be attributed to the surface modification of the nanocrystals, which effectively suppresses ion migration. Our investigations of ion migration activation energy confirmed that the DPPA-NCs exhibit an increased activation energy of 212 meV, higher than the 145 meV for the pristine-NC film (Fig. R3-1, Table R3-1). Besides, the DPPA-NC-based PeLEDs show a lower working temperature than the control devices (Figs. 3k and 3l), which arises from the reduced Joule heat generation. Temperature is a crucial factor influencing ion migration behaviour. An increase in temperature usually accelerates ion migration, compromising device performance.

We have included the ion migration activation energy measurements in our revised manuscript and added discussions (Page 11), which reads “As illustrated in Fig. 3g, the EL spectra of the control device exhibit a redshift accompanied by notable emission broadening, as the operating current densities increase. This phenomenon is attributed to the severe halide ions migration, particularly the different ion migration rates between the Br and I ions under an electric field. In contrast, the EL spectra of the DPPA-NC-based PeLED, which exhibits increased ion migration activation energy (Supplementary Fig. 15 and Supplementary Table 5), maintain a nearly identical shape with a negligible peak shift and slight broadening within 1000 mA cm^{-2} (Fig. 3h). More importantly, this stability persists even at a high current density of 100 mA cm^{-2} for a long time, whereas the EL peak of the device based on pristine-NCs exhibits a noticeable redshift within 2 minutes under similar conditions (Figs. 3i and 3j). Temperature is a crucial factor influencing ion migration behaviour. An increase in temperature usually accelerates ion migration, compromising device performance”.

Fig. R3-1 (Supplementary Fig. 15) | Ion migration activation energy measurement. a–b, The time-dependent currents from 255 K to 305 K by applying bias at 12 V for the (a) pristine-NC film, and (b) DPPA-NC film, respectively. **c–d,** Temperature-dependent plots of $\ln(kT)$ versus $1,000/T$ for the (c) pristine-NC film, and (d) DPPA-NC film, respectively.

Table R3-1 (Supplementary Table 5) | The fitting data from the current decay plots of the pristine-NC film and DPPA-NC film.

Temperature	Pristine-NCs		DPPA-NCs	
	τ_1 (s)	τ_2 (s)	τ_1 (s)	τ_2 (s)
255 K	0.71	45.20	0.73	49.09
260 K	0.71	40.24	0.61	34.95
265 K	0.76	37.43	0.67	29.82
270 K	0.80	33.06	0.75	28.51
275 K	0.83	32.66	0.77	24.34
280 K	0.87	26.8	0.75	20.54
285 K	0.92	26.22	0.81	19.29
290 K	0.95	24.89	0.79	16.78
295 K	0.94	20.62	0.76	13.52
300 K	0.93	21.15	0.75	12.09
305 K	0.86	17.84	0.72	11.43

The method of ion migration activation energy measurement was added in the experimental section (Supplementary Information, page 27):

Ion migration activation energy measurement

The decay in the temperature-dependent temporal response can reflect the kinetics of ionic movement. The temperature-dependent ionic conductivity σ was addressed as²⁰: $\sigma(T) = \sigma_0 \exp(-\frac{E_a}{k_B T})$. Where E_a is the activation energy for ion transport, k_B is the Boltzmann constant.

The decay rate ($k = \tau^{-1}$) of the current decay represents the ionic transport dynamics and is proportional to the ionic conductivity. Therefore, the temperature-dependent decay rates can be used to obtain the thermal active energy for ion migration.

The ion migration activation energy measurement was carried out on the device with a structure of glass/NCs/Au. The NC solution (~15 mg mL⁻¹ in octane) was spun onto the glass at 4000 rpm for 45s. Then the Au interdigital electrode (80 nm) was deposited onto the NC film by thermal evaporation under a high vacuum ($< 5 \times 10^{-4}$ Pa). The samples were placed on a liquid nitrogen thermostat, and Keithley 2635B was used to detect the current signal. The current decay curves were obtained at 12 V and fitted by a double exponential function. The τ_1 is independent of temperature, which relates to the equipment response. The τ_2 represents the time constant of ion migration and is used for calculating the activation energy. The ion migration activation E_a was calculated as the slope of $\ln(kT)-1/T$ using the relation $\ln(kT) = C - \frac{E_a}{k_B T}$, where k can be obtained by $k = \tau^{-1}$ using the time constant τ_2 at different temperatures from 255 K to 305 K.

Secondly, temperature is a crucial factor influencing ion migration behaviour. An increase in temperature typically accelerates ion migration, leading to severe EL spectral shifts. In our work, the DPPA surface modification of NCs exhibits a reduction in nonradiative recombination rates (Fig. 2b, Supplementary Table 1), contributing to reducing heat generation. Besides, the improved conductivity of DPPA-NC film (Figs. 2d-2f) also reduces Joule heat caused by resistance in the PeLED.

In brief, the reduced nonradiative recombination rates, combined with the decreased generation of Joule heat, contribute to lowering the temperature of the working LEDs (Figs. 3k and 3l), thus stabilizing the EL spectra. **Therefore, we acknowledge a correlation between consistent EL spectra and effective thermal management strategies. To clarify the importance of temperature on working PeLED, we also added a sentence in our revised manuscript (Page 11, Paragraph 2), which reads, “More importantly, this stability persists even at a high current density of 100 mA cm⁻² for a long time, whereas the EL peak of the device based on pristine-NCs exhibits a noticeable redshift within 2 minutes under similar conditions (Figs. 3i and 3j). Temperature is a crucial factor influencing ion migration behaviour. An increase in temperature usually accelerates ion migration, compromising device performance.”**

Q3-2. The PLQY of the film significantly impacts the EQE of the device. While the authors reported a near-unity PLQY of DPPA-NCs in solution, what is the PLQY of the perovskite nanocrystal film?

Our revision and responses: The DPPA-NC film presents 90% PLQY, while pristine-NC film only presents below 60%. We have also included the results in our manuscript and described in Page 7, paragraph 2, which reads, “As a result, the photoluminescence quantum yields (PLQYs) improve from 78% of the pristine-NC to near-unity of the DPPA-NC in solution, and from 58% of the pristine-NC to 90% of the DPPA-NC in films.”.

Q3-3. The reviewer noted that the authors fabricated the perovskite film with TMPI. What are the implications of TMPI on both the film properties and device performance?

Our revision and responses: The TMPI molecule was added into the purified NC solution to further passivate the surface halide vacancy of the NCs. As shown in the following figures (Fig. R3-2, Table R3-2, and Fig. R3-3), the TMPI passivated NC showed increased PLQY, and the PeLEDs showed slightly improved device efficiency and EL spectra stability. Nevertheless, there exists a significant performance gap between the DPPA-NCs and pristine-NCs. Therefore,

we conclude that the DPPA surface regulation of the NCs mainly contributes to the improved film properties and device performance.

To clarify the implications of TMPI on both the film properties and device performance, we reorganized the corresponding paragraph in the revised manuscript (page 10), which reads, “We note that the introduced triphenylphosphonium iodide (TMPI) in the PeLED further passivates the NC films, slightly improving the optical properties and enhancing the device performance (Supplementary Fig. 7 and Fig. 14)”.

Fig. R3-2, Table R3-2, and Fig. R3-3 were included as Supplementary Fig. 7, Supplementary Table 2, and Supplementary Fig. 14, respectively.

Fig. R3-2 (Supplementary Fig. 7) | Time-resolved photoluminescence decay curves of the NC films with or without TMPI.

Table R3-2 (Supplementary Table 2) | Summary of time-resolved PL exponential fitting parameters for films of the pristine-NCs and DPPA-NCs with or without TMPI.

Films	f_1 (%)	τ_1 (ns)	f_2 (%)	τ_2 (ns)	τ_{av} (ns)	PLQY (%)	$K_{nr} \times 10^7$ (s ⁻¹)
Pristine-NCs-w/o TMPI	68.39	5.50	31.61	15.56	8.68	52%	5.53
Pristine-NCs+TMPI	70.16	6.44	29.8	20.71	10.69	58%	5.42
DPPA-NCs-w/o TMPI	98.53	10.87	1.47	59.75	11.6	87%	1.12
DPPA-NCs+TMPI	98.20	11.95	1.80	56.45	12.8	90%	0.78

Fig. R3-3 (Supplementary Fig. 14) | Effect of TMPI on device performance. a-c, J-V-L, EQE-L characteristics, and EL spectra stability of the PeLEDs based on the DPPA-NCs+TMPI, and DPPA-NCs-w/o TMPI, respectively. d-f, J-V-L, EQE-L characteristics, and EL spectra stability of the PeLEDs based on the pristine-NCs+TMPI, and pristine-NCs-w/o TMPI, respectively. The TMPI passivates the surface halide vacancy of the NCs, slightly improving the performance of the PeLEDs.

Q3-4. Thermal stability plays a crucial role in both efficiency roll-off and operational stability of the device. What is the thermal stability comparison between pristine perovskite nanocrystals and those treated with DPPA? Providing supporting evidence would strengthen this assessment.

Our revision and responses: The DPPA-NC films demonstrated improved thermal stability compared to pristine-NC films. We assessed the thermal stability of both films, as shown in Fig. R3-4. The photoluminescence (PL) intensity was recorded in situ as the temperature increased. The PL intensity of pristine-NC film decreased at 50 °C and retained only 50% of its initial efficiency when heated to 80 °C. In contrast, the DPPA-NC film showed a slower decline in PL intensity, maintained 80% of its initial efficiency at 80 °C. Additionally, when the NC films were subjected to a hot plate at 80 °C, the DPPA-NC film maintaining PL intensity above 50% of its initial value after 60 minutes of thermal stress, while the PL intensity

of the pristine-NC films dropped below 40% within 20 minutes. **This enhanced thermal stability of the DPPA-NC films is crucial in mitigating efficiency roll-off and ensuring the operational stability of the device.**

Fig. R3-4 (Supplementary Fig. 11) | Thermal stability of the NC films. a, The temperature-dependent normalized PL intensity trajectory of the NC films. **b,** The time-dependent PL intensity trajectory of the NC films at 80 °C.

We have included the results and discussion in the revised manuscript (Page 9), which reads, “We further assessed the thermal stability of these films, as shown in Supplementary Fig. 11. The PL intensity was in situ recorded as the temperature increased. The PL intensity of pristine-NC film decreased at 50 °C and retained only 50% of its initial efficiency when heated to 80 °C. In contrast, the DPPA-NC film showed a slower decline in PL intensity, maintaining 80% of its initial efficiency at 80 °C. Additionally, when the NC films were subjected to a hot plate at 80 °C, the DPPA-NC film retained above 50% of its initial PL intensity after 60 minutes of thermal stress, while the pristine-NC films dropped below 40% within 20 minutes. This enhanced thermal stability of the DPPA-NC films is crucial in mitigating efficiency roll-off and ensuring the operational stability of the device.”. **Fig. R3-4 was also included as Supplementary Fig. 11.**

Q3-5. Considering that thermal management contributes to high brightness and stability, one might expect an improvement in EQE. However, if high thermal conductivity of sapphire substrates helps mitigate efficiency roll-off, it's puzzling that the EQE of sapphire-based devices is lower than that of glass-based ones. What factors might explain this discrepancy?

Our revision and responses: We would like to clarify that there is no inherent difference in the EQE efficiency between devices on sapphire substrates and those on glass substrates. The previous presented slightly lower maximum EQE for sapphire-based devices (EQE: 24.2%) compared to glass-based ones (EQE: 24.8%) is primarily due to the fluctuation of device performance. During the revision, we conducted extensive experiments on sapphire-based devices and performed additional statistical analysis. As shown in Fig. R3-5, the maximum EQE for sapphire-based devices can reach 25.1%, with an average EQE efficiency of 23.1%.

The reason that the EQEs of sapphire-based devices aligns with that of glass-based devices is because the maximum EQEs are reached at low current densities, around 1 mA cm^{-2} . At such low current densities, the devices do not generate significant Joule heat, therefore, the heat dissipation advantage of sapphire substrates does not come into play. Consequently, the thermal conductivity of the substrate does not impact the EQE efficiency at these operating conditions. We have included the discussion in Page 13, Paragraph 1, which reads, “Furthermore, since the peak EQE is attained at lower current densities ($\sim 1 \text{ mA cm}^{-2}$), there is no evident discrepancy in the maximum efficiency between the two PeLEDs (Supplementary Figs. 17 and 18).” Fig. R3-5 was also included as Supplementary Fig. 18. We have also updated the EQE-L curves of sapphire-based devices in Supplementary Fig. 17.

Fig. R3-5 (Supplementary Fig. 18) | Statistical distribution histograms of peak EQE and maximum luminance summarized from 22 DPPA-NC-based LEDs on sapphire substrate.

Supplementary Fig. 17 | EQE-L curves of DPPA-NC-based PeLEDs on the glass or sapphire substrate.

Q3-6. The authors did not do a good job of background checks. To date, tens of reports about thermal management strategies have been employed in perovskite nanocrystal LEDs. The reviewer strongly recommends the authors should summarize previous reports in the Introduction Section to explain the advance of this work.

Our revision and responses: We appreciate the review for pointing out the deficiency in literature review in the introduction section. **We have summarized previous reports on thermal management in perovskite LEDs in the Introduction Section and clarify the motivation of this work, which may explain the advance of our work more straightforwardly.**

Revision in the Introduction Section:

Several thermal management strategies have been developed, encompassing two major approaches: 1) Accelerating thermal dissipation capacity of operating devices by attaching heat spreaders or utilizing high-thermal-conductivity substrates such as silica or sapphire^{12, 17}; and 2) Suppressing the generation of Joule heat by addressing internal factors, such as doping charge-transport layers^{18, 19} and optimizing light extraction nanostructure geometry^{14, 15}. Among them, enhancing the electrical and thermal conductivity of the emissive layers and minimizing nonradiative recombination losses contribute significantly to alleviating the accumulated heat¹².

²⁰. Surface chemistry plays essential roles in determining the conductivity^{21, 22} and regulating surface defects of perovskite materials^{23, 24}, exerting a notable influence on the performance of the devices. Previous investigations have revealed that ligand manipulation can increase the thermal conductivity of nanocrystal (NC) films by 6-10 times²². Inorganic potassium iodide ligands with a high thermal conductivity demonstrated enhanced heat dissipation of NC films, achieving a peak EQE over 23%²⁵. Recently, small-sized aromatic tryptophan ligands were employed to coordinate with CsPb(Br/I)₃ NCs²⁶, exhibiting less-detective surface and superior charge transport properties of the assembled emissive layer, thereby enabling a pure red PeLED with a maximum luminance of 12 910 cd m⁻² and a peak EQE of 22.8%. Nevertheless, it remains an unprecedented challenge to achieve pure red PeLEDs with high efficiency and luminance, suppressed efficiency roll-off, and spectra stability.

In this work, we aim to synergistically suppress Joule heat generation and improve the thermal dissipation of the device to achieve highly bright, efficient, and stable pure-red PeLEDs. Mixed-halide CsPb(Br/I)₃ NCs are considered promising emitters because of the excellent spectra tunability, the efficient radiative recombination, and the limited halide separation in a single particle. Conjugated diphenylphosphoryl azide (DPPA) was employed to regulate the anchoring ligands of CsPb(Br/I)₃ NCs during synthesis, which synergistically enhances the optical properties and carrier transport performance of the NC film. Combined with a thermally conducting sapphire substrate and the pulse driving mode, the overall electroluminescent performance of the NC film was investigated.

REVIEWERS' COMMENTS

Reviewer #1 (Remarks to the Author):

The authors have addressed this reviewer's comments in their response letter and revision.

Reviewer #2 (Remarks to the Author):

The authors have revised the manuscript accordingly, and the paper is ready for publication now.

Reviewer #3 (Remarks to the Author):

Referee's Report

Authors have performed various experiments for supporting their opinion. These results and the responses of revision provide the suitable evidences for publications. Thus, the work will be acceptable for Nature Communications without change.

Reviewer #1:

The authors have addressed this reviewer's comments in their response letter and revision.

We thank the reviewers for their efforts on improving the quality of this manuscript. No Revision is needed.

Reviewer #2:

The authors have revised the manuscript accordingly, and the paper is ready for publication now.

We thank the reviewers for their efforts on improving the quality of this manuscript. No Revision is needed.

Reviewer #3:

Authors have performed various experiments for supporting their opinion.

These results and the responses of revision provide the suitable evidences for publications.

Thus, the work will be acceptable for Nature Communications without change.

We thank the reviewers for their efforts on improving the quality of this manuscript. No Revision is needed.